# A randomised, double-blind, sham-controlled study of left prefrontal cortex 15 Hz repetitive transcranial magnetic stimulation in cocaine consumption and craving

**Francesco Lolli**[1,2☯]*, **Maya Salimova**[1☯], **Maenia Scarpino**[2], **Giovanni Lanzo**[2], **Cesarina Cossu**[2], **Maria Bastianelli**[2], **Brunella Occupati**[3], **Filippo Gori**[1], **Amedeo Del Vecchio**[1], **Anita Ercolini**[1], **Silvia Pascolo**[1], **Virginia Cimino**[1], **Nicolò Meneghin**[1], **Fabio Fierini**[1], **Giulio D'Anna**[1], **Matteo Innocenti**[1], **Andrea Ballerini**[4], **Stefano Pallanti**[1], **Antonello Grippo**[2], **Guido Mannaioni**[1,3]

1 Department of Biomedical, Experimental and Clinical Sciences "Mario Serio", Department of Neuroscience, Psychology, Drug Research and Child Health NEUROFARBA, Department of Health Sciences DSS, Università degli Studi di Firenze, Florence, Italy, 2 Azienda Ospedaliera Universitaria di Careggi, Neurophysiology Unit, Firenze, Italy, 3 Azienda Ospedaliera Universitaria di Careggi, Clinical Toxicology and Poison Control Centre, Firenze, Italy, 4 Azienda Ospedaliera Universitaria di Careggi, Clinical Psychiatry, Firenze, Italy

☯ These authors contributed equally to this work.
* lolli@unifi.it

## Abstract

### Background

Cocaine use disorder (CUD) is a global health issue with no effective treatment. Repetitive Transcranial Magnetic Stimulation (rTMS) is a recently proposed therapy for CUD.

### Methods

We conducted a single-center, randomised, sham-controlled, blinded, parallel-group research with patients randomly allocated to rTMS (15 Hz) or Sham group (1:1) using a computerised block randomisation process. We enrolled 62 of 81 CUD patients in two years. Patients were followed for eight weeks after receiving 15 15 Hz rTMS/sham sessions over the left dorsolateral prefrontal cortex (DLPFC) during the first three weeks of the study. We targeted the DLFPC following the 5 cm method. Cocaine lapses in twice a week urine tests were the primary outcome. The secondary outcomes were craving severity, cocaine use pattern, and psychometric assessments.

### Findings

We randomly allocated patients to either an active rTMS group (32 subjects) or a sham treatment group (30 subjects). Thirteen (42%) and twelve (43.3%) of the subjects in rTMS and sham groups, respectively, completed the full trial regimen, displaying a high dropout rate. Ten/30 (33%) of rTMS-treated patients tested negative for cocaine in urine, in contrast to 4/27 of placebo controls (p = 0.18, odd ratio 2.88, CI 0.9–10). The Kaplan-Meier survival

**Data Availability Statement:** Because data contains potentially identifying and sensitive information, it cannot be shared. The data,

however, will be made available to researchers whose independent review committee has approved the proposed use for meta-analysis. Data will be accessed via the Institutional Poison Control Center of the Azienda Ospedaliera Universitaria Careggi (email: cav@aou-careggi.toscana.it).

**Funding:** Guido Mannaioni—Azienda Ospedaliera Universitaria di Careggi, Fondazione Cassa di Risparmio di Firenze. The funders had no role in study design, data collection and analysis, decision to publish, or preparation of the manuscript.

**Competing interests:** The authors have declared that no competing interests exist.

**Abbreviations:** ASI, Aberrant Salience Inventory; CUD, cocaine use disorder; DLPFC, dorsolateral prefrontal cortex; Hz, Hertz; SUD, substance use disorder; SD, standard deviation; SDQ, Symptoms of Depression Questionnaire; rTMS, repetitive transcranial magnetic stimulation; SHAPS, Snaith Hamilton Pleasure Scale; SUD, substance use disorders; UPPS-P-pers, UPPS-P Impulsive behaviour perseverance subscale; UPPS-P-prem, premeditation subscale; UPPS-P-pos, positive urgency subscale; UPPS-P-neg, negative urgency subscale; UPPS-P-ss, sensation-seeking subscale; VAS, visual analogue scale.

curve did not state a significant change between the treated and sham groups in the time of cocaine urine negativisation (p = 0.20). However, the severity of cocaine-related cues mediated craving (VAS peak) was substantially decreased in the rTMS treated group (p<0.03) after treatment at T1, corresponding to the end of rTMS treatment. Furthermore, in the rTMS and sham groups, self-reported days of cocaine use decreased significantly (p<0.03). Finally, psychometric impulsivity parameters improved in rTMS-treated patients, while depression scales improved in both groups.

## Conclusions

In CUD, rTMS could be a useful tool for lowering cocaine craving and consumption.

## Trial registration

The study number on clinicalTrials.gov is NCT03607591.

## Introduction

Repetitive transcranial magnetic stimulation (rTMS) of the left dorsolateral prefrontal cortex (DLPFC) could be a helpful additional approach to conventional treatment for cocaine use disorder (CUD) [1]. As the addiction persists, dopamine decreases with changes involving the brain's prefrontal region, specifically the orbitofrontal cortex and the cingulate gyrus [2]. Thus, non-invasive brain stimulation techniques may alleviate some of the core symptoms of cocaine abuse [2,3]. rTMS directly impacts several functions altered by cocaine use, including reward, craving, cognitive control, and the influence of dopamine release [4]. However, in terms of clinical outcomes, these responses are diversified. The neurobiological bases of addiction focus on the dopaminergic and glutamatergic systems in modulating transmitter release, metabolism and synaptic function in what is referred as neural plasticity. CUD treatments rely on manipulating these circuits, as well as other inputs and inhibitory systems [4–6].

A rationale for CUD rTMS treatment has occurred in animal studies but with no consensus on humans' correspondences so far [7]. At a low frequency (1 Hz), rTMS is typically inhibitory [8]. At high-frequency, rTMS (>5 Hz) acts as excitatory [9].

To ensure the efficacy of humans' methodology, we need blinded, sham-controlled, prospective, and randomised trials. In the absence of any FDA/EMA-approved pharmacological treatment, international guidelines recommend psychosocial treatment as a first-line approach in CUD. However, it still fails to ensure a long-term response in the majority of patients [10]. A few trials have shown the efficacy of high-frequency rTMS treatment on CUD, but they were mostly open-label, not sham-controlled [1,11–13] and of low power [14].

### Specific objectives or hypotheses

We conducted a randomised, monocentric, sham-controlled, double-blind, parallel-group RCT study, following the protocol previously published [15]. Our specific objectives were to confirm in humans the action of rTMS in reducing cocaine use, craving, depression, anhedonia, and impulsivity, comparing the results of a sham group to an active rTMS unselected typical CUD patients.

## Methods

### Trial design and any changes after trial commencement

The study was a parallel-group, randomised, active-controlled RCT trial with a 1:1 allocation ratio. The study involved the Toxicology Unit, the Neurophysiology Unit, and the Psychiatry Unit of Careggi University Hospital, Florence, Italy. We recruited participants among patients diagnosed with CUD according to DSM-V criteria, seeking CUD treatments and referred to our service either by the Emergency Department or by primary care physicians/Substance Abuse Services in the Florence metropolitan area, Italy. There were no changes after the trial commencement.

### Participants, eligibility criteria, and settings

We assessed the outcomes twice a week and reported results graphically at three significant points: baseline (T0), post-treatment (T1), and eight weeks later (T2). The protocol was approved by the Azienda Ospedaliero-Universitaria Careggi Ethics Committee (CEAVC SPE. 16.309; MagneTox trial, Jul 17, 2017). Following the Helsinki Declaration, all patients signed an informed consent form after receiving the study description. The published method details the inclusion and exclusion criteria stages, rTMS application protocol, and outcomes [15].

The Ethics Committee prohibited the withdrawal of pre-existing pharmacological treatments and psychotherapy. Therefore, psychoactive drugs were either maintained unmodified if chronically administered or titrated/adjusted for steady-state achievement if recently prescribed before starting rTMS. We consequently continued psychotherapy.

### Inclusion and exclusion criteria

The inclusion criteria were age 18–65 years, DSM-5 criteria for CUD, a positive cocaine test in urine, and written informed consent. In addition, a modified pharmacological treatment within 4 weeks, previous rTMS treatment, concomitant alcohol or drug use, a major psychiatric or neurological disorder, illiteracy or cognitive impairment, pregnancy or lactation were all exclusion criteria.

### Procedures, randomisation, and masking

Between October 2017 and December 2019 (2 years), 81 consecutive patients sought treatment for CUD and were eligible, with 62 enrolling. For sex, age, length of CUD, the extent of cocaine use, the 19 patients considered at entry and lost at admission did not vary from the 62 enrolled. Therefore, we assigned patients to either an active rTMS group (32 subjects) or a sham treatment group (30 subjects).

Patient characteristics, basal and after randomisation, are presented in Tables 1 and 2. When two (or more) consecutive negative urine test results occurred at the latest observations, a patient was considered drug-free. The timing considered for the study is detailed in Scarpino [15]. In addition, we conducted craving and psychometric assessments as well as an urine test for cocaine use twice a week. To present the results, baseline (T0) marked the patients' inception, while T1 marked the end of rTMS treatment (four weeks), and T2 indicated the end of follow-up after two months (total twelve weeks).

Sequence generation:

An external team member conducted the randomisation sequence generation using a block randomisation algorithm (1:1).

**Table 1. Patient's data at baseline.**

|  | min | mean | max | SD | n |
|---|---|---|---|---|---|
| VAS base T0 | 0 | 4 | 10 | 3 | 36 |
| VAS peak T0 | 0 | 8 | 10 | 3 | 57 |
| CCQ T0 | 10 | 29 | 69 | 14 | 50 |
| SDQ T0 | 68 | 122 | 198 | 31 | 57 |
| SHPS T0 | 0 | 3 | 12 | 3 | 55 |
| UPPS-neg T0 | 1.1 | 2.0 | 3.6 | 0.5 | 55 |
| UPPS-prem T0 | 1.2 | 2.2 | 3.6 | 0.44 | 56 |
| UPPS-pers T0 | 1.2 | 2.2 | 4.0 | 0.52 | 56 |
| UPPS-ss T0 | 1.3 | 2.2 | 3.8 | 0.64 | 56 |
| UPPS-pos T0 | 1.1 | 2.6 | 4.0 | 0.70 | 56 |
| males n(%) | 49 (83%) |  |  |  |  |
| CUD>10yrs | 35(63%) |  |  |  |  |
| CUD>5yrs | 4(18%) |  |  |  |  |
| CUD<5yrs | 5(19%) |  |  |  |  |
| twice a week use | 25(45%) |  |  |  |  |
| daily use | 17(31%) |  |  |  |  |
| hourly use | 6(11%) |  |  |  |  |
| weekly/monthly use | 7 (15%) |  |  |  |  |
| sniffing | 38 (65%) |  |  |  |  |
| smoking | 22 (38%) |  |  |  |  |
| ev use | 8 (14%) |  |  |  |  |

T0 = basal.

## Blinding, allocation concealment and implementation

Throughout the study, patients, medical staff, and researchers were blinded to the randomisation. In contrast, neurophysiology technicians necessarily could not be blinded but were forbidden from informing other members, including other neurophysiology technicians, medical doctors, and any other person, of the patient allocation. The research was done in a double-blind fashion. There was no control of the fidelity of the blinding.

## Interventions

We randomly assigned CUD patients to either rTMS (treatment group) or sham rTMS (15 sessions of 5 days of treatments and 2 days of rest over three weeks) (control group).

## Transcranial magnetic stimulation

The target point of rTMS was the dorsolateral prefrontal cortex and we employed the 5 cm method. Patients received 15 sessions of high frequency (15 HZ) rTMS with a pulse intensity of 100% (individual threshold levels), 60 pulses per train, an intertrain pause of 15 sec, 40 stimulation trains, and a sum of 2,400 pulses in 13 min.

We used a standard figure-of-eight coil MagPro X100 stimulator (MagVenture, Denmark) and active (code MC-F-B65) and sham coils (MCF-P-B65).

Motor hand hotspot and left DLPFC targeting:

**Table 2. Patients' data after randomisation.**

|  | sham | | | | | rTMS | | | | |
|---|---|---|---|---|---|---|---|---|---|---|
|  | min | mean | max | SD | n | min | mean | max | SD | n |
| VAS base T0 | 0 | 4.5 | 10 | 3 | 30 | 0 | 3.4 | 8 | 3 | 27 |
| VAS base T1 | 0 | 3 | 7 | 3 | 19 | 0 | 2 | 10 | 3 | 17 |
| VAS base T2 | 0 | 2 | 8 | 3 | 11 | 0 | 3 | 9 | 3 | 12 |
| VAS peak T0 | 2 | 9 | 10 | 3 | 27 | 0 | 8 | 10 | 3 | 30 |
| VAS peak T1 | 0 | 7 | 10 | 4 | 17 | 0 | 6 | 10 | 4 | 17 |
| VAS peak T2 | 0 | 6 | 10 | 5 | 11 | 0 | 7 | 10 | 3 | 10 |
| CCQ T0 | 10 | 31 | 69 | 16 | 22 | 10 | 27 | 65 | 13 | 28 |
| CCQ T1 | 0 | 21 | 39 | 13 | 15 | 10 | 22 | 60 | 16 | 17 |
| CCQ T2 | 10 | 28 | 70 | 19 | 12 | 10 | 21 | 67 | 18 | 10 |
| SDQ T0 | 80 | 125 | 188 | 30 | 27 | 68 | 120 | 198 | 32 | 30 |
| SDQ T1 | 64 | 101 | 149 | 24 | 19 | 65 | 104 | 173 | 32 | 19 |
| SDQ T2 | 79 | 110 | 184 | 33 | 15 | 68 | 97 | 173 | 30 | 11 |
| SDQ T0 | 80 | 125 | 188 | 30 | 27 | 68 | 120 | 198 | 32 | 30 |
| SDQ T1 | 64 | 101 | 149 | 24 | 19 | 65 | 104 | 173 | 32 | 19 |
| SDQ T2 | 79 | 110 | 184 | 33 | 15 | 68 | 97 | 173 | 30 | 11 |
| SHAPS T0 | 0 | 4 | 10 | 3 | 26 | 0 | 3 | 12 | 3 | 29 |
| SHAPS T1 | 0 | 2 | 8 | 2 | 19 | 0 | 2 | 9 | 3 | 19 |
| SHAPS T2 | 0 | 3 | 7 | 2 | 15 | 0 | 3 | 11 | 3 | 10 |
| UPPS-neg T0 | 1.4 | 2.1 | 3.3 | 0.4 | 25 | 1.1 | 2.0 | 3.6 | 0.6 | 30 |
| UPPS neg T1 | 1.2 | 2.1 | 3.3 | 0.61 | 19 | 1.10 | 2.43 | 3.8 | .65 | 19 |
| UPPS neg T2 | 1.20 | 2.37 | 3.80 | .62 | 16 | 1.70 | 2.45 | 3.80 | .63 | 12 |
| UPPS-prem T0 | 1.20 | 2.23 | 3.30 | .44 | 26 | 1.40 | 2.24 | 3.30 | .45 | 30 |
| UPPS-prem T1 | 1.00 | 2.12 | 2.80 | .533 | 19 | 1.40 | 2.11 | 3.00 | .47 | 19 |
| UPPS-prem T2 | 1.00 | 2.06 | 2.90 | .454 | 16 | 1.10 | 2.02 | 2.80 | .62 | 12 |
| UPPS-pers T0 | 1.50 | 2.28 | 3.30 | .47 | 26 | 1.20 | 2.28 | 4.00 | .57 | 30 |
| UPPS-pers T1 | 1.20 | 2.02 | 3.00 | .427 | 19 | 1.40 | 2.03 | 3.00 | .47 | 19 |
| UPPS-pers T2 | 1.60 | 2.15 | 2.90 | .48 | 16 | 1.3 | 2.18 | 2.80 | .41 | 12 |
| UPPS-s s T0 | 1.30 | 2.3 | 3.80 | .72 | 26 | 1.4 | 2.4 | 3.4 | .56 | 30 |
| UPPS-ss T1 | 1.30 | 2.42 | 3.60 | .69 | 19 | 1.50 | 2.43 | 3.70 | .592 | 19 |
| UPPS-ss T2 | 1.10 | 2.53 | 4.00 | .86 | 16 | 1.40 | 2.55 | 3.70 | .570 | 12 |
| UPPS-pos T0 | 1.20 | 2.60 | 3.70 | .69 | 26 | 1.10 | 2.66 | 4.0 | .721 | 30 |
| UPPS-pos T1 | 1.00 | 2.56 | 3.80 | .70 | 19 | 1.40 | 2.72 | 4.4 | .80 | 19 |
| UPPS-pos T2 | 1.10 | 2.53 | 4.00 | .75 | 16 | 1.70 | 2.80 | 4.00 | 0.67 | 12 |

T0 = basal.

The theoretical distance between the cortical region being targeted and a reference scalp point established by TMS will be used to identify the target of cortical stimulation (function guided procedure) [16].

The reference point will be the left cortical motor region. The coil will be positioned over the assumed left motor cortex area, and motor evoked potentials (MEPs) collected using the contralateral (right) first dorsal interosseous (FDI) muscle by EMG will be used to determine the hand motor hotspot. The coil will then be adjusted until a position is found where a single-pulse TMS yields repeatable MEPs elicited at the lowest stimulation intensity. The left DLPFC will then be located 5 cm anterior and 2 cm lateral to the hand motor hotspot [17].

The TMS position will be marked on an elastic cap customized for each patient for subsequent rTMS sessions. The coil will be attached to an adjustable arm, and the landmarks on the cap will be checked several times during the rTMS sessions to verify that the coil is correctly positioned on different days. The methods for determining the position of the M1 area and the motor threshold will be repeated before each rTMS session. The intensity of the rTMS will be determined by mapping the FDI muscle response and getting the individual's resting motor threshold (rMT), which indicates the membrane-related excitability of cortical axons.

The rMT will be determined by determining the lowest stimulator output intensity required to obtain 5 out of 10 MEPs greater than 50 V using the MEP technique [18].

We will use a pulse intensity of 100% of their rMT when delivering real rTMS. During the 15 rTMS treatment sessions, the rMT will be measured daily for each participant to ensure safety and efficacy.

The coil center placed at the left DLPFC was pointing 45˚ relative to the midsagittal line. The placebo coil had aspect and sound level identical to the active coil, but the magnetic field was reduced by 80%, although it had a similar cutaneous sensation.

Sample size: The goal was to enroll 60 cocaine-addicted patients over 18 months, which was the calculated sample size for the study's projected power [15].

## Primary outcome

The primary outcome was the time to urine negativisation. The criteria for negativity were two consecutive negative cocaine urine tests at the end of available observation

## Secondary outcomes

The secondary outcomes were VAS, CCQ, and psychometric scales, which indicated craving or changes in psychometric parameters.

## Craving assessment

In this work, we have chosen to use two different scales for craving measurement: the Visual Analog Scale (VAS) and the 10-item Cocaine Craving Questionnaire (CCQ-Brief). For VAS in the respondent's current state, craving ratings are given on a scale of 0 (not at all) to 10 (extremely). VAS scale was applied to measure craving at basal levels (T0), T1, and T2 either under neutral conditions (VAS base) or cocaine use-related cues (VAS peak). During these three-time points, patient's craving in normal daily activities (VAS base) and in cocaine use-related activities (VAS peak) was assessed. In order to distinguish VAS base versus VAS peak, we presented images and recall of cocaine experiences and paraphernalia, and a detailed interview was performed.

Cocaine Craving Questionnaire-Brief was used to assess current craving status ("here and now") at the baseline (T0), T1, and T2 evaluations and twice a week before every programmed urine drug screen test collection designed in the trial protocol for the whole duration of the study (12 weeks). CCQ-brief sums up the ten questions total points, and the final score correlates with craving severity at the moment of CCQ assessment [19]. CCQ is indeed similar to Vas Base, but does not explore the cocaine use related activity (VAS peak). Finally, to build up the swimmer plot analysis, self-reported data on cocaine use were registered at every visit during the study.

## Psychometric scales (behavioural and attitudinal assessment)

The self-reported scales for the psychiatric evaluation included the Symptoms of Depression Questionnaire (SDQ) for depressive symptoms, the Snaith-Hamilton Pleasure Scale (SHAPS) for anhedonia, and the UPPS-P Impulsive Behaviour Scale.

The SDQ is a 44-item instrument with Likert-type answers; the total score is a sum of the items, with higher scores delineating more severe depressive symptoms. The Snaith-Hamilton Pleasure Scale (SHAPS) is a self-reported 14-item instrument assessing anhedonia's presence. Scores $\geq 3$ indicate a state of anhedonia.

The UPPS-P explores five different traits of impulsivity such as i) negative urgency (tendency to act rashly under extreme negative emotions, UPPS-P-neg), ii) lack of premeditation (tendency to act without thinking; UPPS-P-prem), iii) lack of perseverance (inability to remain focused on a task; UPPS-P-pers), iv) sensation seeking (tendency to seek out novel and thrilling experiences; UPPS-P-ss), and v) positive urgency (tendency to act rashly under extreme positive emotions; UPPS-P-pos). Scores of UPPS-P-neg, UPPS-P-ss, and UPPS-P-pos pointing to one outline a higher impulsivity level for these domains. UPPS-P-prem and UPPS-P-pers close to one, on the other hand, imply less severe psychopathology.

**Statistical analysis.** The Shapiro-Wilk test confirmed the non-normal distributions of VAS, CCQ, and psychiatric scales. The Wilcoxon paired test was then employed to test differences in continuous variables (VAS, CCQ, and psychiatric scales) with time. The Kaplan—Meier survival curves plotted the cumulative proportions of drug urine positive patients in the treated and sham groups with time, employing the Mantel-Cox log-rank test to assess the difference. A multivariate Cox model evaluated the influence of different patient characteristics on urine negativisation, with all variables entered with the Wald method. For the time points T1 and T2, the number of cases available at T1 was 41 patients (22 rTMS (70%) and 19 sham (68%)) and at T2 25 patients (13 rTMS (42%) and 12 sham (43%)).

**Trial status.** The study was registered on ClinicalTrials.gov with identifier number NCT03607591. The Enrolment was from October 2017 to April 2020, when the codes were exposed for analysis. The main reason for the delayed registration of the study was the lack of awareness of this policy at the time of the start of the recruitment. We confirm that all ongoing and related trials for this intervention are registered.

## Results

### Participant flow and Baseline data

In two years, 81 patients seeking treatment for CUD were identified as potential study participants, with 62 enrolling (Fig 1 and Table 1). The 19 missed patients did not differ from the 62 enrolled in sex, age, CUD duration, cocaine use frequency, or assumption modalities. The 62 patients who participated in the study (Fig 1) had a mean age of 40,7±9 years; 51 males, 11 females. Patients were randomly assigned to either the active rTMS group (32 subjects) or the sham treatment group (30 subjects) (Fig 1 and Table 2).

### Numbers and analysis for each outcome and subgroup analyses

One and two subjects prematurely disenrolled before receiving treatment in the rTMS AND SHAM group, respectively. Therefore, at the start of the study rTMS/SHAM sessions, 31 patients were treated with rTMS and 28 with sham. A high dropout was observed during the treatments both in rTMS (n = 18; 58%) and sham group (n = 16; 57%). The patients completing the study did not differ from those dropping for frequency and pattern of cocaine use, VAS base or CCQ, but differed for the number of treatments and peak VAS. Dropped outpatients showed an entry average VAS peak of 9.5 (SD 0.56), while patients continuing the study had an entry average VAS peak of 7 (SD 1.8), t-test p<0.006.

Since one patient in the rTMS group and one in the sham group neglected to release urine samples, the statistical analysis of urine drug tests was on 30 and 27 patients in the rTMS and sham group, respectively. When two consecutive drug negative urine tests were

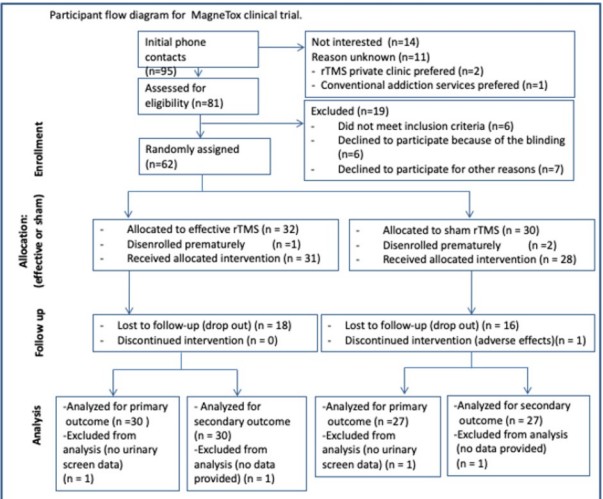

**Fig 1. Participant flow diagram.**

used as criteria for negativity at the end of the observations, 10/30 (33%) of the rTMS treated patients showed urine negativisation, compared to 4/27 (14%) of the sham controls (odd ratio 2.88, CI 0.9–10, p = 0.18). The log-rank test estimated the time to urine negativisation in survival analysis (Fig 2). Four patients entered the study with a negative urine test on the first day of rTMS treatment/sham and did not have another positive test for the study duration. The time to negativisation was listed as 0 for them (immediate). In the survival analysis, the average time to negativisation in the Mantel-Cox model were 90 days (CI 68 to 112 days) and 61 days (CI 40 to 83) in sham and rTMS group, respectively. The Mantel-cox log-rank test wax $X_2 = 1.57$, p = 0.20. The results did not change in a best-case-worst-case scenario including or excluding the four left-censored patients. We estimated the effect of sex, VAS, CCQ, psychiatric scales, drug therapy, psychotherapy, and cocaine use frequency on urine negativisation using a Cox proportional hazard analysis. The model had a significant goodness of fit (p<0.01). and identified VAS peak (cocaine-related cue-induced VAS) at baseline T0 as the only variables associated with urine negativisation (p = 0.037, odd ratio 0.68, CI 0.68–0.98), meaning that higher VAS peak values correspond to a minor effect of sham/rTMS therapy, a higher probability of dropout, and a lower probability of urine negativisation. The swimmer plots graph presents the complete monitoring of the study (Fig 3). Each patient's cocaine use history is colour-coded as red segments for days of self-reported use and green sections for days of self-reported abstinence and is shown in decreasing observation length. Panel A represents the sham subject, while Panel B represents the rTMS treatment group. Urine drug screen results were coded in red, orange, or green dots in each line for positive, borderline, and negative urine drug screens. Triangles marked the drug urine tests at T1 and T2 time points. The top blue bars show the time limits of rTMS or sham application. Self-reported days of cocaine use were 35% and 52% in the rTMS and sham groups, respectively. This percentage differed statistically between groups (Odds ratio for negativisation: 3.4, CI: 1.1 to 10, p<0.03).

At various points during the study, we examined the VAS scale (basal and peak, i.e., neutral cues and cocaine-related cues induced, respectively) and the CCQ, as mean and SD (Fig 4 and Table 2 for raw numbers at baseline (T0), T1, and T2. During the study, while VAS base craving measurement decreased in both treated and sham groups (Wilcoxon test p<0.05 for treated and p<0.02 sham), VAS peak craving showed a significant decrease in

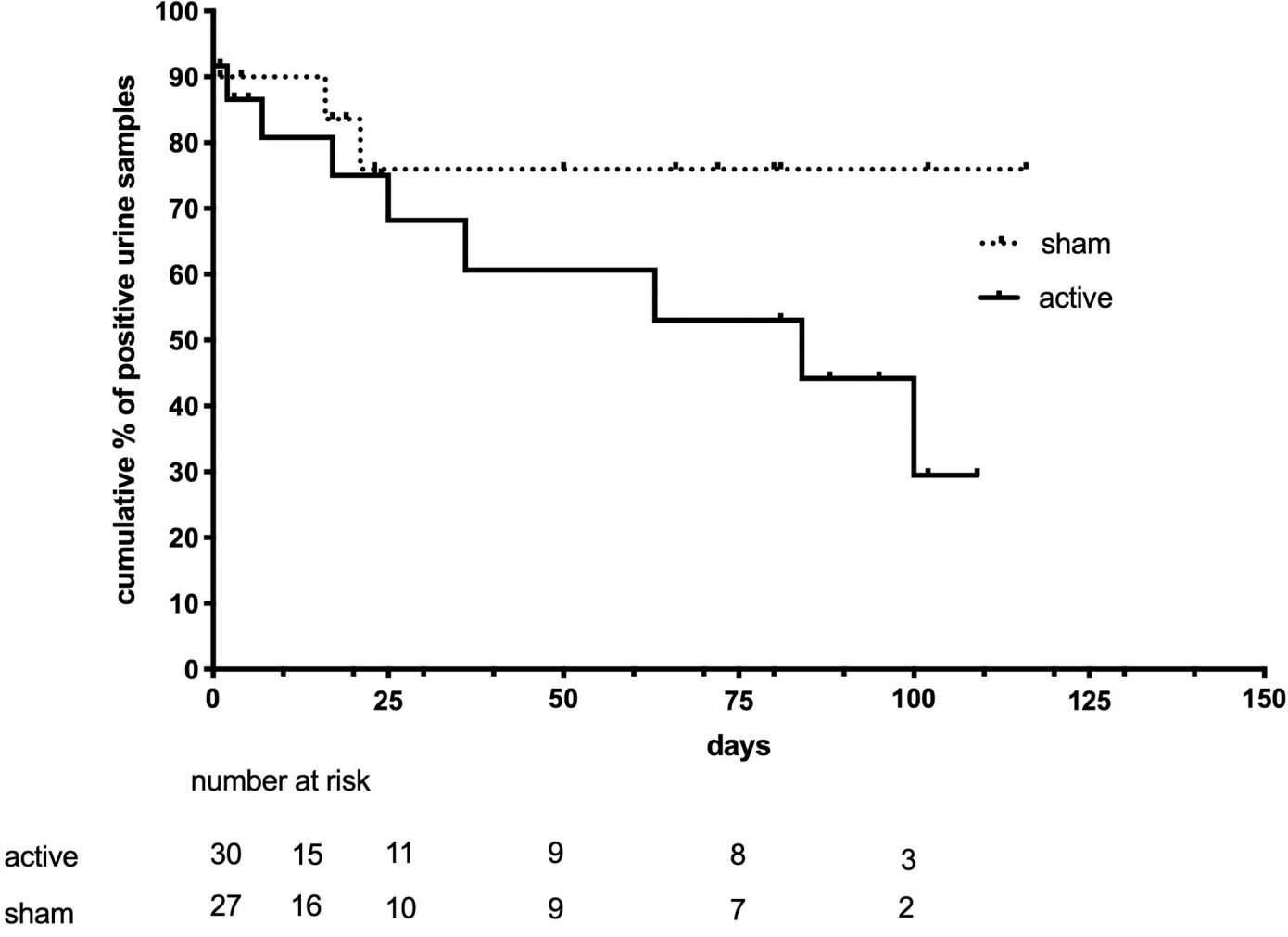

**Fig 2. Cumulative proportion of positive urine samples and numbers at risk in active rTMS and sham-treated group with time (days).** The Mantel-Cox test is p = 0.20. Tics indicate censored patients.

rTMS-treated but not sham-treated patients (p<0.03 T0 vs T1). Only the sham group experienced a significant decrease in CCQ (p<**0.01** T0 vs. T1). Psychiatric evaluation for rTMS and sham group are presented in Fig 5. Compared to baseline, SDQ mean total score was significantly lower at T1 for both treated and sham group (p<0.003 T0 vs T1 and p<0.03 T0 vs T1, respectively). Conversely, the T0-T2 comparisons produced inconclusive results, even though the T1-T2 comparison resulted in a significant elevation of SDQ symptoms only in the sham group (Fig 5, Panel A). The SHAPS mean scores varied significantly in the sham group, with a reduction at T1 as compared to baseline (p<0.03 T0 vs T1). Among the UPPS-P subscales, a considerable amelioration was observed in the negative urgency subscale scores (UPPS-P neg) in the T0-T1 and the T0-T2 comparisons for the rTMS group (Fig 5, panel C, p<0.02 T0 vs T1; p<0.001 T0 vs T2). Conversely, no other differences were outlined by the longitudinal evaluation of other UPPS-P subscales in the rTMS group. Further, none of the UPPS-P subscales delineated conclusive and stable longitudinal variations in the sham group, except for a reduction in the perseverance subscale (UPPS-P pers) (Fig 5, panel D, p<0.02 T0 vs T1).

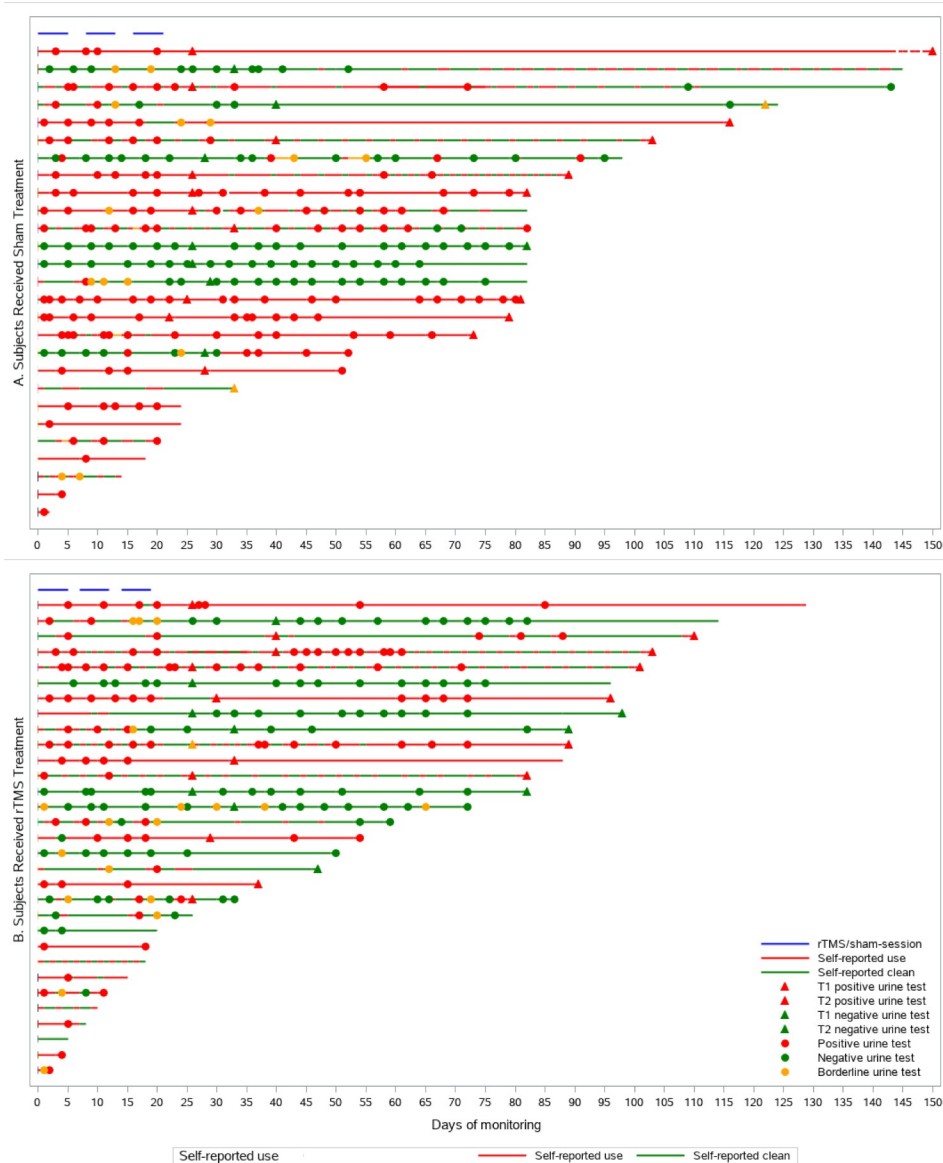

**Fig 3. Patients' histories are presented in decreasing length of observation and colour coded as red for the days of use or referred use and green for abstinence days.** Panel A represents the subject randomised to the sham treatment group, while panel B represents the active rTMS treatment. We marked each line's urine drug screen results, codified as red, orange, or green dots for positive, borderline, and negative urine drug screens. Triangles marked the urine drug screen tests at T1 and T2 time points. The top blue bars indicate the application of rTMS or sham treatment.

## Harms

Only a minor treatment-related adverse effect was observed in a single patient undergoing one sham treatment session and experienced mild and transient paraesthesia.

## Discussion

Our study reports the improvements (self-reported use, VAS peak and UPPS-P-neg rTMS vs sham) in patients with CUD following rTMS treatment stimulating the left DLPFC. Moreover, although not statistically different, urine negativisation time was improved in

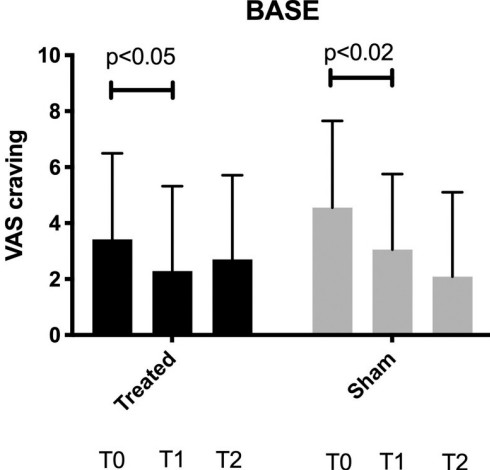

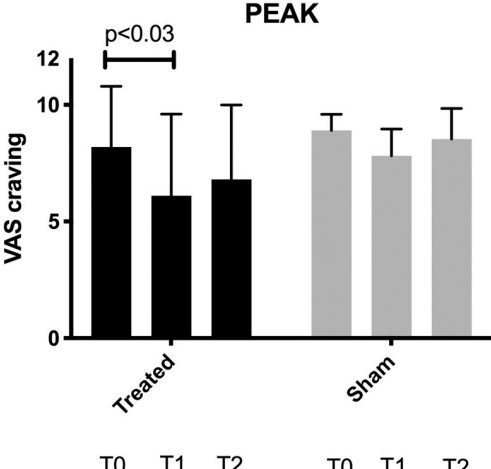

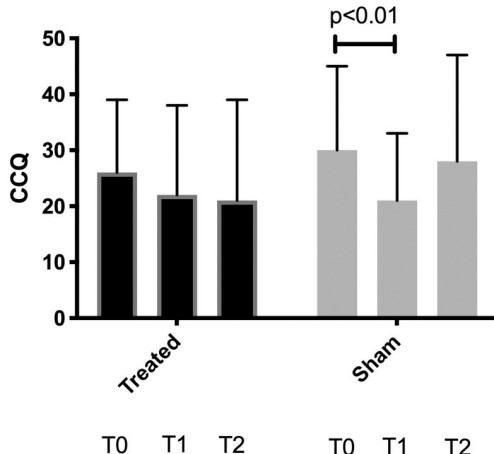

**Fig 4. Craving scales.** VAS (basal and peak) and CCQ (mean and SD) in the rTMS treated and sham-treated groups at T0, T1, and T2. The Wilcoxon signed-rank test evaluated differences.

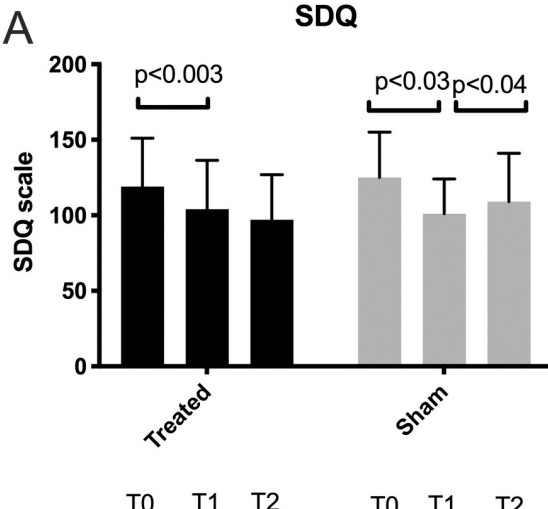

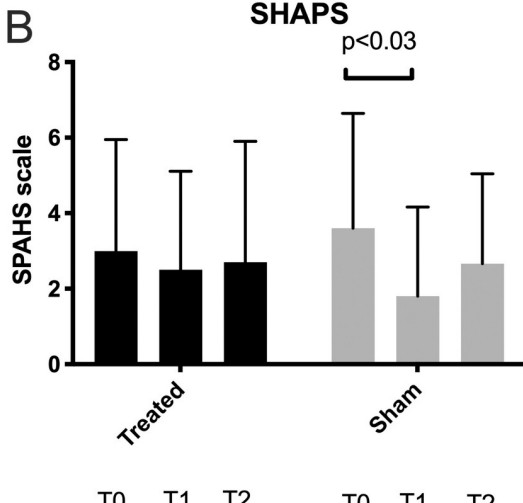

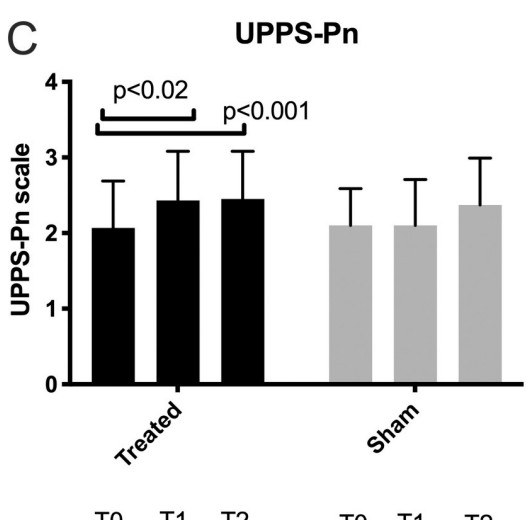

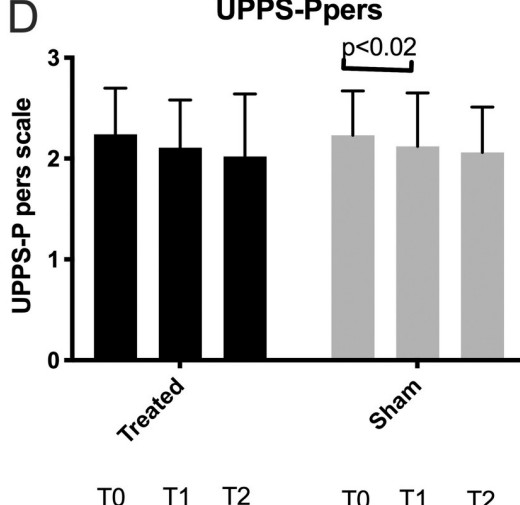

**Fig 5. SDQ (panel A), SHAPS (panel B), UPPS-Pn (panel C), UPPS-P-pers (panel D) in the rTMS and sham-treated groups at T0, T1, and T2.** Significant differences were calculated with the Wilcoxon signed-rank test.

the rTMS population. Interestingly, the self-reported cocaine use was statistically different between the rTMS and sham groups. These results are among the firsts conducted in a large cohort of patients with the sham-controlled procedure, with an adequate follow-up, and in an actual real clinical setting without a preliminary selection of patients, thus avoiding selection biases. A previously published paper described our study methodology [15]. We aimed to adopt a sham-controlled, double-blind design to control the expected placebo effect and fine-tune the different aspects of the rTMS effect in CUD abstinence and craving.

The neurological basis for rTMS's disease-fighting effects is unknown. The state of the cortex, age, sex, genetics, neurotransmitter and receptor differences, connectivity of the stimulated region, and finally, the position of the coil relative to the target population and head geometry are all likely to play a role with the stimulation protocol.

RCT with sham control is the standard gold method for evaluating rTMS clinical efficacy in other disorders, such as depression [20]. Previous clinical research in double-blinded sham-controlled randomised studies have been published [21,22]. These older studies underlined either a late reduction (over three months following rTMS application) in cocaine intake measured in hair analysis [21] or cocaine versus money choice in the rTMS group [22]. However, both these studies showed either a limited number of recruited patients or lower rTMS frequency or did not perform any follow-up. A recent RCT study had similar characteristics to our study in stimulus frequency, size of stimulation (mostly 5 cm method) and intensity, with significant results in diminished self-reported cocaine use and craving. Similarly to our study, no difference in cocaine urine levels was observed [23].

On the other hand, all the remaining studies were not double-blinded, nor sham-controlled or randomised [1,11–13]. Interestingly, these studies have underlined the effectiveness of rTMS in craving control [1,11,12,14,24], and two studies which measured cocaine urinalysis showed a better outcome in the rTMS group [14,25]. Nonetheless, these older studies showed insufficient quality and lacked adequate power in rTMS enrolled patients and other limitations such as the sample size/treatment duration [11], missing controls, and low numbers of cases.

Our study missed the neuronavigation system, which has been described as potentially useful even if its superiority has not been definitively proven so far [20]. The standard anatomical method (5 cm method) employed in our study can be easily generalised. Its efficacy has been demonstrated in depression, in which rTMS is now considered a proven therapeutic intervention. Indeed, recent rTMS application guidelines indicate the F4/F3 sites at the 10–20 EEG System (the "Beam F3" algorithm) as possibly a more anatomically accurate non-navigated method for targeting the DLPFC in the major depression trials [20] but there is no definite proof [6].

As unexpected results, we registered a high dropout rate in randomised patients throughout the study in both arms, not infrequent in CUD studies [26]. Indeed, the original sample size calculation was determined to be 30 in each arm to provide 80% power, with a 2-sided significance level of $\alpha = 0.05$, to detect a difference for clinical improvement, assuming a dropout rate of 20% of the patients. Unfortunately, the dropout rate in the real setting was higher (nearly 60%), which is probably one reason for the proposed primary outcome failure. Moreover, the urine sample collection is scarcely adequate in the proposed study design. Indeed, urine sample collections were extremely erratic throughout the study in both arms, which reduced the study power. However, it is interesting to notice that this is a common problem in studies involving substance use disorder population s, often bypassed by using statistical strategies to handle the missing data. For this reason, in outpatient settings, urine sample collections seem not to be a reliable biomarker for a clinical trial's goals. A standardised, reliable biomarker investigating for cocaine consumption should be necessarily proposed in future studies.

There was no statistically significant difference in the time to urine negativisation between patients who received active rTMS and those who received a sham treatment in our trial. However, the survival curves did initially differentiate between the groups, and the average time to negativisation was shorter with the active therapy. The frequency of patients with urine negativisation was significantly higher in the same group, indicating an early effect of rTMS in this experimental setting and probably confirming the need for a subchronic rTMS treatment [25].

Positive results were found by using a swimmer plot analysis following an interview about daily cocaine use. We chose this type of analysis since it reflects a better picture of the cocaine pattern of use reported day by day, similarly to Madeo and co-workers [25]. Interestingly, the results are very comparable to the most recent RCT rTMS/CUD report [23], and both study confirm the same pattern of results.

Analysing the secondary outcomes revealed useful information on the specific effect on VAS, craving, depression, and impulsivity, which corresponded to symptoms release. CUD

often clusters with impulsive behaviour and depression, with a complex interaction between these psychopathological areas [27,28]. Craving is a hallmark symptom of CUD [2,19]. VAS for cocaine craving assessment has been previously validated in CUD patients [29]. Previous studies demonstrated that rTMS to the left DLPFC was able to reduce VAS cue-induced craving in long-term heroin users [30] as well as in a case report of opioid and cocaine use disorder [31]. However, pre-post change analysis for VAS cocaine cue-induced craving was not performed due to time limitations for each visit [32]. Visual Analogue Scale was used to measure global craving for substances. Indeed, the association between the intensity of craving and substance use disorders severity is well established [27]. Therefore, high VAS at peak in CUD patients suggests that higher cocaine-related cues induced craving could be proposed as a negative prognostic factor for the success of CUD treatment in general and rTMS treatment. For this reason, we propose to differentiate craving measurement in basal and peak measurement.

The VAS scale application has already been proposed in pain [33], as well as in addiction research [11]. However, even the effect on VAS is not permanent and seems to fade shortly after the end of rTMS treatment. This is consistent with the recent introduction of rTMS protocols that implies the repetition of the rTMS sessions, reaching for the maintenance of the result on cocaine use in the long term [25].

As for depression, anhedonia, and impulsivity, both substance-induced and independent depressive symptoms can contribute to relapse into cocaine utilisation [34]. In this sense, monitoring these dimensions of psychiatric interest is of utmost clinical value, and an open-label study already suggested a potential effect of rTMS on cocaine users' psychopathology [35,36]. SDQ scores showed similar results in the T0-T1 interval for both the rTMS and the sham group. This concurrent variation may partly be explained by a significant placebo effect, extensively acknowledged in rTMS treatments. Besides, rapid-onset and transient amelioration of depressive symptoms have already been described in treating mood disorders [27,28]. Conversely, even though T0-T2 comparisons led to inconclusive findings, a significant potential for the rTMS group could be tied to different T1-T2 trends. Indeed, a further, non-significant improvement of SDQ scores was observed in the rTMS group, whereas a loss of early progress was observed in the sham group. The presented data's inconclusive effect is most likely due to the small sample size and high dropout rate. Nonetheless, the role of mood in subjects undergoing rTMS for substance use disorders warrants further investigation, as it had prognostic value in an RCT involving methamphetamine-dependent patients [37].

Concerning anhedonia, the present trial did not show relevant improvements of this psychopathological dimension, except for a single comparison in the sham group (pT0-T1 < 0.03). However, the significance of this finding is unclear since anhedonia is a trans-nosographic dimension [38].

A previous study showed an overall negligible effect of rTMS on impulsive behaviours [39], and the present trial generally confirmed this observation. More in detail, sensation seeking, lack of premeditation, and positive emotion management were globally stable across time. Regarding perseverance, a similar trend (although non-significant for the rTMS group) was observed for both the treated and the sham group, and it is probably to be ascribed to early improvements and behavioural consequences of the above-mentioned placebo effect. On the contrary, significant variations were observed for the treated group negative urgency scores with clinically and theoretically sound explanations [40]. Thus, negative emotional affect, which was represented in the overall sample (as proved by the depression and anhedonia baseline scores), is the main drive of cocaine use. In this sense, it can be speculated that positive toxicological outcomes are indirectly tied to improvements in this specific sub-domain, rather than compulsive and sensation-seeking behaviour.

Finally, this study and the recent RCT by Garza-Villareal and coworkers had similar clinical and rTMS parameters, both as frequency of stimulation (5 Hz versus 10Hz), and number, intensity, and site of stimulations (mostly 5cm method) [23]. The similar results support the external validity of the findings.

## Limitations and generalizability

This study has limitations that may have hampered its effectiveness. First, the high dropout rate underpowered the results. Furthermore, a longer rTMS treatment duration, as well as an extended follow-up period, may allow for the detection of longer-term differences. The randomisation protocol and the rigorous analysis, which was limited to pre-specified criteria, eliminated potential bias sources reinforcing the results.

## Conclusion

Our final interpretation should weigh the benefits against the risks, with the therapy being well tolerated and free of side effects. Considering all the evidence obtained in the study, such as the effect on self-reported use, depression, VAS at peak, and impulsivity, rTMS could be considered beneficial and safe add-on therapy in CUD.

## Supporting information

**S1 Checklist.**
(DOC)

## Acknowledgments

We thank the patients and care providers who offered invaluable support. We thank Mr Leonardo Casini for graphical support.

## Author Contributions

**Conceptualization:** Francesco Lolli, Maya Salimova, Giovanni Lanzo, Andrea Ballerini, Stefano Pallanti, Antonello Grippo, Guido Mannaioni.

**Data curation:** Francesco Lolli, Maya Salimova, Giovanni Lanzo, Cesarina Cossu, Maria Bastianelli, Brunella Occupati, Filippo Gori, Amedeo Del Vecchio, Anita Ercolini, Silvia Pascolo, Virginia Cimino, Nicolò Meneghin, Fabio Fierini, Giulio D'Anna, Matteo Innocenti, Andrea Ballerini, Stefano Pallanti, Antonello Grippo, Guido Mannaioni.

**Formal analysis:** Francesco Lolli, Maya Salimova, Giovanni Lanzo, Cesarina Cossu, Maria Bastianelli, Amedeo Del Vecchio, Anita Ercolini, Silvia Pascolo, Virginia Cimino, Nicolò Meneghin, Fabio Fierini, Giulio D'Anna, Matteo Innocenti, Andrea Ballerini, Stefano Pallanti, Guido Mannaioni.

**Funding acquisition:** Francesco Lolli, Maya Salimova, Guido Mannaioni.

**Investigation:** Francesco Lolli, Maya Salimova, Maenia Scarpino, Giovanni Lanzo, Cesarina Cossu, Maria Bastianelli, Brunella Occupati, Filippo Gori, Amedeo Del Vecchio, Anita Ercolini, Silvia Pascolo, Virginia Cimino, Nicolò Meneghin, Fabio Fierini, Giulio D'Anna, Matteo Innocenti, Andrea Ballerini, Stefano Pallanti, Antonello Grippo, Guido Mannaioni.

**Methodology:** Francesco Lolli, Maya Salimova, Maenia Scarpino, Giovanni Lanzo, Brunella Occupati, Filippo Gori, Silvia Pascolo, Virginia Cimino, Nicolò Meneghin, Giulio D'Anna, Matteo Innocenti, Andrea Ballerini, Stefano Pallanti, Antonello Grippo, Guido Mannaioni.

**Project administration:** Guido Mannaioni.

**Resources:** Francesco Lolli, Maya Salimova, Guido Mannaioni.

**Software:** Maya Salimova, Maenia Scarpino, Stefano Pallanti.

**Supervision:** Francesco Lolli, Maya Salimova, Maenia Scarpino, Giovanni Lanzo, Cesarina Cossu, Maria Bastianelli, Brunella Occupati, Fabio Fierini, Andrea Ballerini, Stefano Pallanti, Antonello Grippo, Guido Mannaioni.

**Validation:** Maya Salimova, Maenia Scarpino, Cesarina Cossu, Maria Bastianelli, Brunella Occupati, Filippo Gori, Amedeo Del Vecchio, Anita Ercolini, Silvia Pascolo, Virginia Cimino, Nicolò Meneghin, Fabio Fierini, Giulio D'Anna, Matteo Innocenti, Andrea Ballerini, Stefano Pallanti, Antonello Grippo, Guido Mannaioni.

**Visualization:** Francesco Lolli, Maya Salimova, Maenia Scarpino, Giovanni Lanzo, Brunella Occupati, Filippo Gori, Amedeo Del Vecchio, Anita Ercolini, Silvia Pascolo, Virginia Cimino, Nicolò Meneghin, Fabio Fierini, Giulio D'Anna, Matteo Innocenti, Andrea Ballerini, Stefano Pallanti, Antonello Grippo, Guido Mannaioni.

**Writing – original draft:** Francesco Lolli, Maya Salimova, Maenia Scarpino, Giovanni Lanzo, Brunella Occupati, Andrea Ballerini, Stefano Pallanti, Antonello Grippo, Guido Mannaioni.

**Writing – review & editing:** Maya Salimova, Guido Mannaioni.

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
