## [Decision Letter · Decision Letter 0]

26 Jul 2021

PONE-D-21-19751

A randomized, double-blind, sham-controlled study of left prefrontal cortex 15 Hz repetitive transcranial magnetic stimulation in cocaine consumption and craving.

PLOS ONE

Dear Dr. Lolli,

Thank you for submitting your manuscript to PLOS ONE. After careful consideration, we feel that it has merit but does not fully meet PLOS ONE’s publication criteria as it currently stands. Therefore, we invite you to submit a revised version of the manuscript that addresses the points raised during the review process.

We look forward to receiving your revised manuscript.

Kind regards,

Bernard Le Foll, M.D., Ph.D.

Academic Editor

PLOS ONE

Journal Requirements:

2. Thank you for submitting your clinical trial to PLOS ONE and for providing the name of the registry and the registration number. The information in the registry entry suggests that your trial was registered after patient recruitment began. PLOS ONE strongly encourages authors to register all trials before recruiting the first participant in a study.

1) your reasons for your delay in registering this study (after enrolment of participants started);

2) confirmation that all related trials are registered by stating: “The authors confirm that all ongoing and related trials for this drug/intervention are registered”.

4. Thank you for stating the following in the Funding Section of your manuscript:

“Azienda Ospedaliera Universitaria Careggi, Firenze, Italy; Fondazione Cassa di Risparmio Firenze, Universita’ degli Studi di Firenze”

We note that you have provided additional information within the Funding Section that is not currently declared in your Funding Statement. Please note that funding information should not appear in the Acknowledgments section or other areas of your manuscript. We will only publish funding information present in the Funding Statement section of the online submission form.

 “Guido Mannaioni. Azienda Ospedaliera Universitaria di Careggi, Fondazione Cassa di Risparmio di Firenze.

6. Please ensure that you refer to Figure 4 in your text as, if accepted, production will need this reference to link the reader to the figure.

Reviewers' comments:

Reviewer's Responses to Questions

**Comments to the Author**

1. Is the manuscript technically sound, and do the data support the conclusions?

Reviewer #1: Partly

Reviewer #2: Yes

Reviewer #3: Yes

2. Has the statistical analysis been performed appropriately and rigorously? 

Reviewer #1: No

Reviewer #2: Yes

Reviewer #3: Yes

3. Have the authors made all data underlying the findings in their manuscript fully available?

Reviewer #1: Yes

Reviewer #2: Yes

Reviewer #3: Yes

4. Is the manuscript presented in an intelligible fashion and written in standard English?

Reviewer #1: Yes

Reviewer #2: Yes

Reviewer #3: No

5. Review Comments to the Author

Reviewer #1: The objective of this single-center, parallel group, randomized (sham) controlled trial (RCT) is to assess the effectiveness rTMS therapy for treating cocaine use disorder (CUD). The study was registered as a RCT within the clinicaltrials.gov registry (with a legit NCT number), and was approved by the respective IRB/Ethics Committee. While the study objectives sound interesting, is important, and on target, a number of shortcomings were observed, in regards to abiding by the CONSORT guidelines for conducting and reporting results of high-quality randomized controlled trials (RCTs). Some other (statistical) comments were also added.

1. Methods:

Methods reporting require an orderly manner following CONSORT guidelines, without repeating information, such as Trial Design, Participant Eligibility criteria and settings, Interventions, Outcomes, sample size/power considerations, Interim analysis and stopping rules. Randomization (details on random number generation, allocation concealment, implementation), and Blinding considerations should be mentioned explicitly. The authors are advised to create separate subsections for each of the possible topics (whichever necessary), and that way produce a very clear writeup. I see the Authors already made a sincere attempt; however, they are advised to write it carefully, following nice examples in the manuscript below:

https://www.sciencedirect.com/science/article/pii/S0889540619300010

Specific comments below:

(a) For instance, the randomization and allocation concealment should be made very clear (they are NOT the same thing); the trial staff recruiting patients should NOT have the randomization list. Randomization should be prepared by the trial statistician, and he/she would not participate in the recruiting.

(b) Sample size/power: There is no sample size/power paragraph presented; it is also not clear whether sample size determinations were done using the primary outcome variable (time to urine negativization). Also, sample size calculations should consider the desired effect size under consideration.

(c) Statistical Analysis:

(i) For the survival analytic endpoint, how is the (right) censoring determined? Is it administrative censoring, or something else?

(ii) Fig 2 is not really a Kaplan-Meier plot; one needs to plot the survival curves for urine negativization corresponding to the "sham" and "active" groups, and then conduct a log-rank test to produce the desired p-value.

(iii) The fit of the multivariate Cox model should be accompanied by necessary goodness-of-fit assessments, and checking the proportional hazards assumptions, through popular tests.

2. Results:

(a) The authors should check that any statement of significance should be followed by a p-value in the entire Results section. Otherwise, the Results section look adequate followed by a detailed discussion.

Reviewer #2: The paper by Lolli et al describes the effect of rTMS treatment on cocaine treatment-seeking cocaine addicts. The paper is well written and well designed representing yet another experimental effort that ultimately supports the use of rTMS in the treatment of cocaine addiction. Analysis of experimental data is now more accurate than a previous version of the paper I had seen elsewhere.

I would simply encourage the authors to comment on the possible neurobiological basis for TMS effects.

As such the discussion is merely clinical and reflections/ideas about the the neurobiology underlying clinical effects would help in reducing the 'exoteric aura' around TMS.

Reviewer #3: Lolli and colleagues report on a single site, randomized, double-blind, parallel-group, randomized controlled trial of high frequency rTMS vs. sham for cocaine use disorder. They describe strengths in their trial design and large sample size, however note their challenges with retention of participants as drop-out rates were high (equally in both groups). They did not find any significant differences in their primary outcome of urine toxicology, however a number of important secondary measures did show superiority of rTMS, including outcomes related to self reported use, depression, certain indices of craving, and impulsivity. Importantly, they note that rTMS was well tolerated without any significant adverse effects.

Given the growing interest in neuromodulation for addictive disorders, this study is a welcome addition to the literature. The authors are correct in stating that the vast majority of currently published studies are far too small and under-powered in nature to be confident about clinically significant therapeutic benefit. Given the unique and substantial risks in managing severe substance using populations, it also needs to be demonstrated in clinical trials that these treatments that require intensive follow up (e.g. daily treatment visits required of rTMS) can be feasible in those that abuse substances. This may be particularly challenging for very destabilizing substances such as cocaine. This manuscript provides data to address these current shortcomings in the literature well.

The main significant criticism is that this manuscript fails to address is the issue of rTMS targeting, which has become a field of intense debate and study. Current approaches typically include MRI based anatomical targeting, fMRI-based connectivity targeting, or other brain biomarkers (e.g. EEG). The rationale and excitement for neuromodulation in the addictions field is the ability to more specifically target aberrant neurocircuitry. Thus, the current paper’s target of “PMC/DLPFC” is very imprecise according to current standards. The PMC and DLPFC are relatively disparate regions of the cortex. Standard rTMS procedures, even those using only scalp-based measurements, aim to target only the DLPFC. One such method is the “Beam F3 method” which the uses the 10-20 EEG placement system is likely more precise. The authors do cite the Beam F4 method in their discussion, stating it is the most anatomically accurate non-MRI navigated method, but do not seem to use it themselves.

Other comments I have are relatively minor, but there are many of them, mostly related to increasing clarity of the writing of the manuscript. It will be important for the authors to address issues like blinding and statistical analyses to account for missing data, as these relate to the methodological issues that are important to the paper.

Introduction

- Line 61-62 is too definitive of a statement for such preliminary research cited.

- Life 65 don’t recommend wording of “restoring symptoms”, should be treating or alleviating symptoms

- Line 66-67 needs citations.

Methods

- Please consider reporting your protocol according to the CONSORT checklist of information to include when reporting a randomised trial.

- For clarity, I would suggest listing inclusion criteria and exclusion criteria separately

- For clarify, I would recommend using a different word than biweekly. It is a little bit unclear since this word can mean every 2 weeks or it can mean twice a week.

- Line 119: suggest using terminology, thus T0 should be “baseline” as previously described in the page before.

- Line 123: What does a “neurophysiology technician” refer to?

- Blinding was reported for participants and medical operators. Please comment on blinding by the raters for standardized scales. Was blinding successful and were any measures conducted to assess for fidelity of blinding? As mentioned above, it is not clear who the “neurophysiology technicians” are, but it is mentioned that they were not blinded – does this affect the integrity of the results in any way?

- I recommend using standard terminology for describing the motor threshold method. It is described in a confusing way, with it first referenced as an “individual threshold level” on line 126, then there is some vague description of MEPs around line 134-135 (is this using EMG? Was this resting motor threshold? Then it is mentioned again later in line 141-142 where this is more of a discussion based point about what the motor threshold represents. This can all be consolidated into a few lines or a paragraph all together.

- Please explain the rationale for targeting of the premotor cortex. It was not mentioned in the introduction or the beginning of the methods and was named for the first time on line 129 with just the acronym.

- The description of the target site and landmarking is difficult to follow. Line 132 to 134 seem to suggest there is landmarking done, but it is not clear what these landmarks are, when later it seems to suggest the landmarking is just based on uniform measurements (regardless of individual head shape and size). Where does this landmarking protocol come from? Has it been standardized or referenced before?

- It is not clear why the stimulation sits is “PMC/DLPFC”. These are relatively different sites on the cortex, and it is rather imprecise to lump these together. Furthermore, in the Abstract, the target site is only described as “DLPFC” alone.

- Line 154-156 is better reserved for an introduction or discussion section.

- The VAS protocol needs to be better described. It is not clear how the “VAS base” and “cocaine use-related activities” was actually conducted. Those descriptions provided (e.g. “meeting people who consume the drug”) do not explain whether these are imagined, or whether they are based on image or video cues as is standard in this field. I understand that the citations provided may better describe it, but a brief version of the steps should be available within the paper.

- For the Cocaine Craving Questionnaire, it is not clear how this “here and now” assessment is any different than the VAS. The VAS as currently described also sounds like it is a current craving assessment.

- For the self reported data on cocaine use, was there any specific method used? For example the Timeline Follow Back Method? Was quantity of use collected?

- Line 171 – the word “relevant” symptoms meant to say more severe total burden of symptoms? Does the scale differentiate between different types of symptoms in terms of importance and how would this be described in a total score?

- Please reword line 180-181, it is too difficult to understand in this form

- Statistical analysis: please describe how missing data was accounted for, particularly as your primary outcome is negativity at the end of available observation, may have biased the results towards an effect by missing participants that dropped out due to relapse.

Results:

- For the differences stated between the drop outs and those that continued the study, are those measures listed in line 207-208 referring to baseline cocaine use or throughout the study?

- Were the drop out percentages listed referring to drop out before T1 or T2? Did this affect the analysis?

Discussion:

- Beam F3 may have been more accurate but this is not the way the authors did it in this study

- The discussion of dropouts and difficulty obtaining urine samples is a very interesting paragraph. The authors propose alternate biomarkers of cocaine consumption. However, would an alternate biomarker have any influence on the drop out rate?How much is the drop out rate related to the issues with urine collection? In other words, is it a significant problem that participants that do not drop out yet still a urine sample cannot be obtained?

- The authors discuss using a daily interview and the swimmer’s plot as a good way of identifying daily use. Is there a reason why the Timeline Followback Method was not used? The authors cite Garza-Villarreal et al (2021) who did indeed use the TLFB.

- In describing the strengths of this study as being large and a blinded RCT, the authors do not credit other such studies including Garza-Villarreal et al. (2021), who similarly present a rigorous study design. It may be interesting to include a brief comparison with this study in the discussion section. Similarly, there are other large, well designed RCTs published in methamphetamine use disorder, which although is not exactly the same as CUD, has obvious overlap. It may be interesting to have some discussion of these studies as well in relation to this manuscript.

6. PLOS authors have the option to publish the peer review history of their article (what does this mean?). If published, this will include your full peer review and any attached files.

Reviewer #1: No

Reviewer #2: **Yes: **Marco Diana

Reviewer #3: No

---

## [Author Response · Author response to Decision Letter 0]

29 Sep 2021

Firenze 29-09-2021

Response to the Editor and referees of our article PONE-D-21-19751 "A randomised, double-blind, sham-controlled study of left prefrontal cortex 15 Hz repetitive transcranial magnetic stimulation in cocaine consumption and craving."

Dear Editor of Plos one,

 Thank you for the decision letter dated July 26, 2021, and the reviewers' comments to improve the paper. The manuscript has been revised in response to the reviewers' suggestions and editorial guidelines. Please find a point-by-point response to the journal requirements and the Reviewers' comments in our resubmission.

We hope that the revised version of the paper will now meet the publication criteria of PLOS ONE.

Sincerely,

Francesco Lolli, MD, Ph.D.

Comments for the Editor

A rebuttal letter (this file) is included below and named 'Response to reviewers'. A revised 'Manuscript with Track Changes' is included as well as an unmarked clean copy 'Manuscript.'

Our updated financial statement in the cover letter is

"“Azienda Ospedaliera Universitaria Careggi, Firenze, Italy; Fondazione Cassa di Risparmio Firenze, Italy”

 We employed the guidelines and software recommended for checking figures.

Our study protocol was preliminarily published in Scarpino et al. 2019.

Our manuscript meets the 'PLOS ONE's style requirements, including those for file naming.

We endorse the immediate and preliminary study registration. Unfortunately, we experienced a delay in the study registration, now detailed in the text. 

As per the journal's editorial policy, we included in the Methods section of our paper:

1) After checking recent literature (M Al-Durra, BMJ 2020; 369:m982[Prospective registration and reporting of trial number in randomised clinical trials: global cross-sectional study of the adoption of ICMJE and Declaration of Helsinki recommendations]), a phrase was added in the methods section as requested ("The main reason for delayed registration was lack of awareness of this policy at the time of starting recruiting"). 

2) We confirmed that all related trials are registered by stating: "The authors confirm that all ongoing and related trials for this intervention are registered".

3) the 'Funding Information' and 'Financial Disclosure' sections do now match.

Our amended financial statement is “Azienda Ospedaliera Universitaria Careggi, Firenze, Italy; Fondazione Cassa di Risparmio Firenze, Italy”.

Furthermore, we corrected it in the online system.

5.

The data availability statement is now introduced as: “Data cannot be shared publicly because contain sensible information. The data will be available to investigators whose independent review committee has approved the proposed use for meta-analysis.”

6. we refer to Figure 4 in the text.

7. there is no supporting/supplementary information

Comment to reviewer#1

Original comment:"The objective of this single-centre, parallel-group, randomised (sham) controlled trial (RCT) is to assess the effectiveness rTMS therapy for treating cocaine use disorder (CUD). The study was registered as an RCT within the clinicaltrials.gov registry (with a legit NCT number), and was approved by the respective IRB/Ethics Committee. While the study objectives sound interesting, is important, and on target, a number of shortcomings were observed, in regards to abiding by the CONSORT guidelines for conducting and reporting results of high-quality randomised controlled trials (RCTs).

Methods reporting require an orderly manner following CONSORT guidelines, without repeating information, such as Trial Design, Participant Eligibility criteria and settings, Interventions, Outcomes, sample size/power considerations, Interim analysis and stopping rules. Randomisation (details on random number generation, allocation concealment, implementation), and Blinding considerations should be mentioned explicitly. The authors are advised to create separate subsections for each of the possible topics (whichever necessary), and that way produce a very clear writeup. I see the Authors already made a sincere attempt; however, they are advised to write it carefully, following nice examples in the manuscript below: (Erbe et al 2019)"

After our efforts to comply with space limitations, the reviewer rightly reports shortcomings in abiding by the CONSORT guidelines for conducting and reporting results of high-quality randomised controlled trials (RCTs). As suggested, we now applied stringently the guidelines such as Trial Design, Participant Eligibility criteria and settings, Interventions, Outcomes, sample size/power considerations, Randomisation (details on random number generation, allocation concealment, implementation), and Blinding considerations. In addition, we used subsections and the format signalled in Erbe et al. 2019, as suggested.

________

Original comment:""(a) For instance, the randomisation and allocation concealment should be made very clear (they are NOT the same thing); the trial staff recruiting patients should NOT have the randomisation list. Randomisation should be prepared by the trial statistician, and he/she would not participate in the recruiting.

(b) Sample size/power: There is no sample size/power paragraph presented; it is also not clear whether sample size determinations were done using the primary outcome variable (time to urine negativisation). Also, sample size calculations should consider the desired effect size under consideration.

c) Statistical Analysis"

i) For the survival analytic endpoint, how is the (right) censoring determined? Is it administrative censoring, or something else?

(ii) Fig 2 is not really a Kaplan-Meier plot; one needs to plot the survival curves for urine negativisation corresponding to the "sham" and "active" groups, and then conduct a log-rank test to produce the desired p-value.

(iii) The fit of the multivariate Cox model should be accompanied by necessary goodness-of-fit assessments, and checking the proportional hazards assumptions, through popular tests.

(a) The randomisation and allocation concealment and blinding were specified in the appropriate sections

(b) The sample size calculation was described in the method paper by Scarpino et al. (2019). They are also added in the present study as suggested

(c)

i. It is an administrative censoring. All subjects complete the course of the study and are known to have experienced either of the two outcomes at the end of the study

ii. We corrected fig.2 to a typical Kaplan-Meyer plot as suggested

iii. The goodness of fit and p values of model are now reported as suggested

Original comment:"The authors should check that any statement of significance should be followed by a p-value in the entire Results section

(a) in the results section, any statement of significance is now followed by a p-value

Reviewer#2

Original comment:""I would simply encourage the authors to comment on the possible neurobiological basis for TMS effects".

 We thank the reviewer for the suggestion. We considered better the neurobiological bases for rTMS effect, now presented in the introduction and discussed in relation to our results

Reviewer#3

"

Original comment:"The main significant criticism is that this manuscript fails to address is the issue of rTMS targeting, which has become a field of intense debate and study. Current approaches typically include MRI based anatomical targeting, fMRI-based connectivity targeting, or other brain biomarkers (e.g. EEG). The rationale and excitement for neuromodulation in the addictions field is the ability to more specifically target aberrant neurocircuitry. Thus, the current paper's target of "PMC/DLPFC" is very imprecise according to current standards. The PMC and DLPFC are relatively disparate regions of the cortex. Standard rTMS procedures, even those using only scalp-based measurements, aim to target only the DLPFC. One such method is the "Beam F3 method" which the uses the 10-20 EEG placement system is likely more precise.

We do agree with the referee that rTMS targeting has become a hot topic for discussion and research. We are aware of the numerous possible targets. Our idea was to conform to the most popular target, and the name "PMC/DLPFC" came from a discussion with an international referee in our previous pubblication (Scarpino et al, 2019). At the time of protocol design, the beam F3 methods were not preferred. We now simplifyed the target name as simply DLPFC as in the main literature and following referee suggestion.

Original comment:"Introduction

- Line 61-62 is too definitive of a statement for such preliminary research cited.

- Life 65 don't recommend wording of "restoring symptoms", should be treating or alleviating symptoms

- Line 66-67 needs citations.

Line 61-62 was changed accordingly

Life 65 now read "alleviating symptoms."

Line 66-67 have a new citation

Methods

Original comment:"- Please consider reporting your protocol according to the CONSORT checklist of information to include when reporting a randomised trial.

- For clarity, I would suggest listing inclusion criteria and exclusion criteria separately

- For clarify, I would recommend using a different word than biweekly. It is a little bit unclear since this word can mean every 2 weeks or it can mean twice a week.

- Line 119: suggest using terminology, thus T0 should be "baseline" as previously described in the page before.

- Line 123: What does a "neurophysiology technician" refer to?

- Blinding was reported for participants and medical operators. Please comment on blinding by the raters for standardised scales. Was blinding successful and were any measures conducted to assess for fidelity of blinding? As mentioned above, it is not clear who the neurophysiology technicians" are, but it is mentioned that they were not blinded – does this affect the integrity of the results in any way?

We employed the CONSORT checklist for reporting a randomised trial according to this comment and those from reviewer#1.

Inclusion and exclusion criteria were listed separately

Biweekly was written twice a week.

Line 119 T0 was corrected to "baseline (T0)"

In line 123, please refer to https://college.mayo.edu/academics/health-sciences-education/clinical-neurophysiology-technology-program-minnesota/

Blinding. We now discuss the blinding for all personell. There was no control for the fidelity of the blinding. The neurophysiology technician has no contact with the raters, doctors, other patients, or other technicians, and it is now stated. At the time of starting of the study, there was no possibility to blind the technician to the two different coils. Nonetheless, we believe this has no bearing on the outcome of the study.

Original comment:"I recommend using standard terminology for describing the motor threshold method. It is described in a confusing way, with it first referenced as an "individual threshold level" on line 126, then there is some vague description of MEPs around line 134-135 (is this using EMG? Was this resting motor threshold? Then it is mentioned again later in line 141-142 where this is more of a discussion based point about what the motor threshold represents. This can all be consolidated into a few lines or a paragraph all together.

- Please explain the rationale for targeting of the premotor cortex. It was not mentioned in the introduction or the beginning of the methods and was named for the first time on line 129 with just the acronym.

- The description of the target site and landmarking is difficult to follow. Line 132 to 134 seem to suggest there is landmarking done, but it is not clear what these landmarks are, when later it seems to suggest the landmarking is just based on uniform measurements (regardless of individual head shape and size). Where does this landmarking protocol come from? Has it been standardised or referenced before?

- It is not clear why the stimulation sits is "PMC/DLPFC". These are relatively different sites on the cortex, and it is rather imprecise to lump these together. Furthermore, in the Abstract, the target site is only described as "DLPFC" alone.

- Line 154-156 is better reserved for an introduction or discussion section.

We now used standard terminology for describing the motor threshold method, and the whole procedure is described in a new paragraph.

The targeting of the premotor cortex is now described at the beginning of the methods and named.

The description of the target site and landmarking is new. We detailed the protocol and how it was done and standardised, and referenced

The "PMC/DLPFC" is now identified at the beginning of the methods as DLPFC through the paper. Therefore, the target site in the abstract conforms to this name.

Line 154-156 is now part of the discussion.

Original comment:"The VAS protocol needs to be better described. It is not clear how the "VAS base" and "cocaine use-related activities" was actually conducted. Those descriptions provided (e.g. "meeting people who consume the drug") do not explain whether these are imagined, or whether they are based on image or video cues as is standard in this field. I understand that the citations provided may better describe it, but a brief version of the steps should be available within the paper.

- For the Cocaine Craving Questionnaire, it is not clear how this "here and now" assessment is any different than the VAS. The VAS as currently described also sounds like it is a current craving assessment.

- For the self reported data on cocaine use, was there any specific method used? For example the Timeline Follow Back Method? Was quantity of use collected?

The methods “VAS base" and "cocaine use-related activities" were better described, along with what was done, as suggested.

The distinctions between the Cocaine Craving Questionnaire and the VAS were clarified. CCQ is indeed similar to VAS Base, but does not explore the cocaine use related activity (VAS peak). 

For the self-reported data, there was a quantity of use collected defined as single day of use. We did not employ any more specific method, since the TMB was not of general use at the time of the beginning of the study.

Original comment:"Line 171 – the word "relevant" symptoms meant to say more severe total burden of symptoms? Does the scale differentiate between different types of symptoms in terms of importance and how would this be described in a total score?

- Please reword line 180-181, it is too difficult to understand in this form

- Statistical analysis: please describe how missing data was accounted for, particularly as your primary outcome is negativity at the end of available observation, may have biased the results towards an effect by missing participants that dropped out due to relapse.

Line 171, it is “more severe symptoms”. The scale is a score composite with more item. The total score we employed is a a global evaluation

Line 180-181 was rewritten

- Statistical analysis: after the randomisation procedure, the number of missing values are balanced between the group. See the number of exposed in the Kaplan-Meyer plot

Original comment:"For the differences stated between the drop outs and those that continued the study, are those measures listed in line 207-208 referring to baseline cocaine use or throughout the study?

- Were the drop out percentages listed referring to drop out before T1 or T2? Did this affect the analysis?

We are refering to basal cocaine use. 

The differences stated between the dropouts and those that continued the complete study in lines 207-208 refer to baseline cocaine use, with the idea of a possible future stratification of cases at entry.

The drop out percentages listed refer to observed at any time before termination of the study, they are listed in table 2 and text (numbers in results). Thus, drop out are distributed all along with the investigation.

Original comment:"Beam F3 may have been more accurate but this is not the way the authors did it in this study

- The discussion of dropouts and difficulty obtaining urine samples is a very interesting paragraph. The authors propose alternate biomarkers of cocaine consumption. However, would an alternate biomarker have any influence on the drop out rate?How much is the drop out rate related to the issues with urine collection? In other words, is it a significant problem that participants that do not drop out yet still a urine sample cannot be obtained?

- The authors discuss using a daily interview and the swimmer's plot as a good way of identifying daily use. Is there a reason why the Timeline Followback Method was not used? The authors cite Garza-Villarreal et al (2021) who did indeed use the TLFB.

- In describing the strengths of this study as being large and a blinded RCT, the authors do not credit other such studies including Garza-Villarreal et al. (2021), who similarly present a rigorous study design. It may be interesting to include a brief comparison with this study in the discussion section. Similarly, there are other large, well designed RCTs published in methamphetamine use disorder, which although is not exactly the same as CUD, has obvious overlap. It may be interesting to have some discussion of these studies as well in relation to this manuscript.

Beam F3 method could have improved the results, but we employed a more commonly used target.

Comment on dropouts and difficulty obtaining urine samples. The dropout from the study end the urine collection in all patients. The randomization balance the drop out between the 2 arms In order to discuss the importance of new biomarkers we do believe they cannot influence the drop-out rate. Regarding the explanation on how the patients in tretment still have diffculties in urine sample collection, unfortunately we do non have any explanation. Indee it is well know that CUD and substance use disorders populations, is a very difficult population to study. We pointed ou these difficulties in the discussion section. (Pani et al. 2011)

We are discussing the Beam F3 method in the setting of our study and others, the role of dropouts, TLFB analysis versus the swimmer plot. 

The swimmer plot is the method of reference to display individual patients results. As mentioned to the previous referee for craving, we did not employ any more specific method, since TLFB was not of general use at the time of the beginning of the study.

Finally, we discuss the critical points raised and identified by Garza-Villarreal et al. (2021) versus our results. 

We recognise the relevance of the study from Garza-Villareal (2021), whis is now confronted, described and compared to our study in the discussion. We believe that the result, very simiilar, reinforce the conclusions.

The methamphetamine use disorder are cited.

Sincerely

Prof. Francesco Lolli, MD, PhD

Università di Firenze, Italy

Francesco.lolli@unifi.it

---

## [Decision Letter · Decision Letter 1]

28 Oct 2021

A randomized, double-blind, sham-controlled study of left prefrontal cortex 15 Hz repetitive transcranial magnetic stimulation in cocaine consumption and craving.

PONE-D-21-19751R1

Dear Dr. Lolli,

We’re pleased to inform you that your manuscript has been judged scientifically suitable for publication and will be formally accepted for publication once it meets all outstanding technical requirements.

Kind regards,

Bernard Le Foll, M.D., Ph.D.

Academic Editor

PLOS ONE

Additional Editor Comments (optional):

Reviewers' comments:

Reviewer's Responses to Questions

**Comments to the Author**

1. If the authors have adequately addressed your comments raised in a previous round of review and you feel that this manuscript is now acceptable for publication, you may indicate that here to bypass the “Comments to the Author” section, enter your conflict of interest statement in the “Confidential to Editor” section, and submit your "Accept" recommendation.

Reviewer #1: All comments have been addressed

Reviewer #2: All comments have been addressed

Reviewer #3: All comments have been addressed

2. Is the manuscript technically sound, and do the data support the conclusions?

Reviewer #1: (No Response)

Reviewer #2: (No Response)

Reviewer #3: Yes

3. Has the statistical analysis been performed appropriately and rigorously? 

Reviewer #1: (No Response)

Reviewer #2: (No Response)

Reviewer #3: Yes

4. Have the authors made all data underlying the findings in their manuscript fully available?

Reviewer #1: (No Response)

Reviewer #2: (No Response)

Reviewer #3: Yes

5. Is the manuscript presented in an intelligible fashion and written in standard English?

Reviewer #1: (No Response)

Reviewer #2: (No Response)

Reviewer #3: Yes

6. Review Comments to the Author

Reviewer #1: (No Response)

Reviewer #2: (No Response)

Reviewer #3: (No Response)

7. PLOS authors have the option to publish the peer review history of their article (what does this mean?). If published, this will include your full peer review and any attached files.

Reviewer #1: No

Reviewer #2: **Yes: **Marco Diana

Reviewer #3: No

---

## [Editor Report · Acceptance letter]

5 Nov 2021

PONE-D-21-19751R1 

A randomised, double-blind, sham-controlled study of left prefrontal cortex 15 Hz repetitive transcranial magnetic stimulation in cocaine consumption and craving. 

Dear Dr. Lolli:

I'm pleased to inform you that your manuscript has been deemed suitable for publication in PLOS ONE. Congratulations! Your manuscript is now with our production department. 

Kind regards, 

on behalf of

Dr. Bernard Le Foll 

Academic Editor

PLOS ONE